# Improvements of Flexural Properties and Thermal Performance in Thin Geopolymer Based on Fly Ash and Ladle Furnace Slag Using Borax Decahydrates

**DOI:** 10.3390/ma15124178

**Published:** 2022-06-13

**Authors:** Ng Yong-Sing, Liew Yun-Ming, Heah Cheng-Yong, Mohd Mustafa Al Bakri Abdullah, Phakkhananan Pakawanit, Petrica Vizureanu, Mohd Suhaimi Khalid, Ng Hui-Teng, Hang Yong-Jie, Marcin Nabiałek, Paweł Pietrusiewicz, Sebastian Garus, Wojciech Sochacki, Agata Śliwa

**Affiliations:** 1Centre of Excellence Geopolymer and Green Technology (CEGeoGTech), Universiti Malaysia Perlis (UniMAP), Kangar 01000, Malaysia; nicholas.zai.1130@gmail.com (N.Y.-S.); cyheah@unimap.edu.my (H.C.-Y.); mustafa_albakri@unimap.edu.my (M.M.A.B.A.); venessa42@live.com (N.H.-T.); hangyongjie0919@gmail.com (H.Y.-J.); 2Faculty of Chemical Engineering Technology, Universiti Malaysia Perlis (UniMAP), Kangar 01000, Malaysia; 3Faculty of Mechanical Engineering Technology, Universiti Malaysia Perlis (UniMAP), Kangar 01000, Malaysia; 4Synchrotron Light Research Institute, 111 University Avenue, Muang District, Nakhon Ratchasima 30000, Thailand; phakkhananan@slri.or.th; 5Faculty of Materials Science and Engineering, Gheorghe Asachi Technical University of Iasi, 61 D. Mangeron Blvd., 700050 Iasi, Romania; 6Technical Sciences Academy of Romania, Dacia Blvd 26, 030167 Bucharest, Romania; 7Chemical Services and Environment (Technical Support) Section, Sultan Azlan Shah Power Station, TNB Janamanjung Sdn Bhd, Seri Manjung 32040, Malaysia; suhaimik@tnb.com.my; 8Department of Physics, Częstochowa University of Technology, 42-201 Częstochowa, Poland; marcin.nabialek@pcz.pl (M.N.); pawel.pietrusiewicz@pcz.pl (P.P.); 9Faculty of Mechanical Engineering and Computer Science, Częstochowa University of Technology, 42-201 Częstochowa, Poland; sebastian.garus@pcz.pl (S.G.); wojciech.sochacki@pcz.pl (W.S.); 10Division of Materials Processing Technology and Computer Techniques in Materials Science, Silesian University of Technology, 44-100 Gliwice, Poland; agata.sliwa@polsl.pl

**Keywords:** thin geopolymer, borax decahydrate, flexural properties, thermal performance

## Abstract

This paper elucidates the influence of borax decahydrate addition on the flexural and thermal properties of 10 mm thin fly ash/ladle furnace slag (FAS) geopolymers. The borax decahydrate (2, 4, 6, and 8 wt.%) was incorporated to produce FAB geopolymers. Heat treatment was applied with temperature ranges of 300 °C, 600 °C, 900 °C, 1000 °C and 1100 °C. Unexposed FAB geopolymers experienced a drop in strength due to a looser matrix with higher porosity. However, borax decahydrate inclusion significantly enhanced the flexural performance of thin geopolymers after heating. FAB2 and FAB8 geopolymers reported higher flexural strength of 26.5 MPa and 47.8 MPa, respectively, at 1000 °C as compared to FAS geopolymers (24.1 MPa at 1100 °C). The molten B_2_O_3_ provided an adhesive medium to assemble the aluminosilicates, improving the interparticle connectivity which led to a drastic strength increment. Moreover, the borax addition reduced the glass transition temperature, forming more refractory crystalline phases at lower temperatures. This induced a significant strength increment in FAB geopolymers with a factor of 3.6 for FAB8 at 900 °C, and 4.0 factor for FAB2 at 1000 °C, respectively. Comparatively, FAS geopolymers only achieved 3.1 factor in strength increment at 1100 °C. This proved that borax decahydrate could be utilized in the high strength development of thin geopolymers.

## 1. Introduction

Geopolymers are amorphous inorganic binders made up of a three-dimensional network, formed via the alkali activation of aluminosilicates [1]. The aluminosilicates are materials that are rich in silica and alumina and are the primary requirement in forming the main framework of Si–O–Al in a geopolymer structure [2]. Natural clays such as kaolin [3] and metakaolin [4] or industrial by-products, such as fly ash [5] and slag [6] are the common aluminosilicate sources used in the formation of geopolymers. Fly ash (FA), which is primarily composed of spherical glassy particles, is typically produced by the combustion of bituminous coal and exhibits pozzolanic properties [7]. Ladle furnace slag (LFS), which is a type of steel slag, possesses cementitious characteristics, which are suitable to be applied in geopolymer synthesis [8]. However, unlike ground granulated blast furnace slag (GGBFS), LFS is less explored in the geopolymer formation.

The three-dimensional framework of SiO_4_ and AlO_4_ provides geopolymer remarkable physical and mechanical properties due to its ceramic-like characteristics [9]. In fact, geopolymers have excellent thermal properties, and extended heating could facilitate the extent of geopolymerization, yielding the formation of a geopolymer matrix, thereby enhancing the mechanical strength of the geopolymer [10]. Furthermore, according to past literature, the thermomechanical properties of geopolymers could be enhanced using additives [5], fiber reinforcement [11,12] and blended geopolymers with the incorporation of two or more precursor aluminosilicates [13,14,15]. For instance, more complex crystalline phases were formed in blended geopolymers at high temperatures, which was beneficial to the high-strength development. Indeed, our previous published work, [16], also demonstrated that FA/LFS-blended geopolymers had better flexural and thermal performance than FA geopolymers.

In recent years, researchers have attempted to develop new types of geopolymers, by substituting the silica or alumina parts with other materials such as phosphates, germanates and borates to produce phosphate-based geopolymers [17], aluminogermanate geopolymers [18] and boroaluminosilicate geopolymers [19,20,21,22] with adequate performances. For instance, Williams and van Riessen [19] produced borosilicate geopolymers using silica fume and alkali activator mixed with NaOH solution and anhydrous borax and achieved 57 MPa of compressive strength. Nazari et al. [20,21] and Bagheri et al. [22] produced boroaluminosilicate geopolymers using anhydrous borax as an alkali activator and reported an improvement in flexural and compressive strength due to the formation of B–O bonds. Moreover, Liu et al. [23] and Dupuy et al. [24] reported anhydrous borax and borax decahydrate showed retarding effects in improving the setting time of fly ash geopolymers and metakaolin geopolymers, respectively.

Thus far, the effect of borax addition on the strength development of geopolymers at elevated temperatures is not known. Borax could be used as flux at high temperatures, which was used in ceramic sintering [25,26]. Borax would decompose after being subjected to heat and produce boron oxide (B_2_O_3_) that forms a glassy adhesive medium and integrates the pyrolysis remnants from the composites, initiating the phase transition at a lower sintering temperature [27]. Nevertheless, geopolymers experienced phase transformation at a high temperature of ~1000 °C [15,28]. It is crucial to evaluate that the incorporation of borax might facilitate the phase transformation of geopolymers at lower elevated temperatures, which is beneficial to the development of high-flexural strength crystalline body.

As a result, this brings the focus on the potential of utilizing borax in thin geopolymer to induce high strength properties. Studies regarding the incorporation of borax in the field of geopolymers at high temperatures are still limited. This paper proposes the incorporation of borax decahydrate as an additive into the synthesis of 10 mm thin geopolymers based on fly ash and ladle furnace slag. The addition of borax decahydrate is expected to improve the flexural properties of geopolymers by lowering the glass transition temperature, which could increase the degree of crystallization at elevated temperature exposure. The flexural properties and thermal performance of thin geopolymers were comprehensively discussed in terms of morphology, phase, and functional groups. The pore structure and distribution are evaluated using Synchrotron radiation X-ray tomographic microscopy and the correlation between porosity and flexural properties of thin geopolymers is also elucidated.

## 2. Methodology

### 2.1. Materials

Class F FA and LFS are used as the aluminosilicate sources in this study. The sources of these materials are from Sultan Azlan Shah Power Station, Seri Manjung, Perak, Malaysia, and Southern Steel Berhad, Penang, Malaysia, respectively. The oxides components of FA and LFS determined using X-ray Fluorescence (XRF) and the loss on ignition (LOI) evaluated using the mass difference from 29 °C to 950 °C are tabulated in Table 1. FA was mainly made up of silica (56.30 wt.%) and alumina (28.00 wt.%) while the major oxides of LFS were calcium oxide (63.59 wt.%) and silica (21.30 wt.%). The particle size distribution of FA and LFS was displayed in Figure 1. FA had 80.2% of particles size finer than 65 μm with d_10_ = 36.3 μm, d_50_ = 52.6 μm and d_90_ = 76.5 μm whereas 74.3% of LFS particles size was finer than 65 μm with d_10_ = 35.6 μm, d_50_ = 55.2 μm and d_90_ = 81.0 μm. Figure 2 (Figure 2a,b) shows the microstructure of FA with spherical shapes and LFS with irregular shapes of plate-like structure.

The aluminosilicates were alkali-activated by the mixture of liquid sodium silicate (Na_2_SiO_3_) and sodium hydroxide (NaOH) solution. The liquid Na_2_SiO_3_ supplied by South Pacific Chemicals Industries Sdn. Bhd., Selangor, Malaysia contained 30.1 wt.% SiO_2_, 9.4 wt.% of Na_2_O and 60.5 wt.% of H_2_O with specific gravity of 1.4 g/cm^3^ and viscosity of 0.4 Pa s, respectively, at 20 °C. The 97% assay of caustic soda pellets (HmbG^®^ Chemicals, Sigma-Alrich, Taufkirchen, Germany) was mixed with distilled water to prepare NaOH solution. Sodium tetraborate decahydrate (Na_2_B_4_O_7_·10H_2_O) in powder form was used as the borax additive in this study. The borax decahydrate (Analytical Reagent Grade, Fisher Scientific, Leicestershire, UK) with particle size of 50 mesh (300 μm) and density of 1.73 g/cm^3^. The borax decahydrates particles were clustered and aggregated in certain areas (Figure 2c).

### 2.2. Synthesis of Thin FA/LFS Geopolymers

The thin FA/LFS blended geopolymers were synthesized by mixing FA and LFS with alkali activator using aluminosilicate/activator ratio of 2.5. The weight ratio of FA and LFS was fixed at 60:40. The alkali activator was prepared by mixing 12 M NaOH solution with liquid Na_2_SiO_3_ at Na_2_SiO_3_/NaOH ratio of 4.0. The mixing compositions used were based on our previously reported work [29]. The borax decahydrate was added at 2 wt.%, 4 wt.%, 6 wt.% and 8 wt.% into the geopolymer slurry and stirred until homogenous paste was achieved. Then, the paste was cast and compacted into molds (160 × 40 × 10 mm). An oven curing was performed on the molded geopolymers at 60 °C for 6 h, followed by a 24 h curing step at room temperature (RT). After that, the thin geopolymers were demolded and kept at RT for 28 days. The thin FA/LFS geopolymer without the addition of borax decahydrate was labeled as FAS. The thin geopolymers added with borax decahydrate (FAB) were denoted as FAB2, FAB4, FAB6 and FAB8, respectively, in accordance with the weight percentage of borax addition.

### 2.3. Elevated Temperature Exposure

The thin geopolymers were exposed to elevated temperatures in a muffle furnace after 28-days curing. The heating rate was set at 3 °C/min with 2 h soaking time. Heat was applied to FAS geopolymers at 300 °C, 600 °C, 900 °C, 1000 °C, and 1100 °C, and to FAB geopolymers at 300 °C, 600 °C, 900 °C, and 1000 °C. The elevated temperatures were set based on the thermo-physical and chemical changes of geopolymers from past literature [28] whereby reactions such as further geopolymerization, dehydration and crystallization of geopolymers could be observed within these temperature ranges. The upper limit of heating temperature for FAS and FAB geopolymers was set at 1100 °C and 1000 °C, respectively, as melting of thin geopolymers occurred above these temperatures (Figure 3). For comparison purposes, a set of unexposed geopolymers was kept.

### 2.4. Testing and Analysis

Mass and dimension of thin geopolymers before and after exposure to evaluated temperatures were determined to evaluate the bulk density according to BS EN 12390-7. The apparent porosity and water absorption measurements were investigated using ASTM C642. Wet mass (*M_w_*), oven-dried mass (*M_d_*) and suspended mass (*M_s_*) were determined and used to calculate the apparent porosity and water absorption of thin geopolymers as shown in Equations (1) and (2), respectively.
(1)$$\mathrm{Apparent}\;\mathrm{Porosity}=\frac{M_{w}-M_{d}}{M_{w}-M_{s}}$$
(2)$$\mathrm{Water}\;\mathrm{Absorption}=\frac{M_{w}-M_{d}}{M_{d}}$$

The flexural strength of thin geopolymers was evaluated based on ASTM C348 using standard three-point-bending test operated with Instron Machine Series 5569 Mechanical Tester. The loading rate was fixed at 1 mm/min and span length of 110 mm. Data from five samples from each set of thin geopolymer were collected to calculate the average flexural strength.

The microstructural analysis was assessed using JEOL JSM-6460LA model scanning electron microscope (SEM). A thin layer of platinum coating was applied to the fracture surface which had been cut into small pieces prior to the analysis to avoid electrostatic charging when taking the SEM images.

NETZSCH STA 449 Jupiter was performed to evaluate the thermal analysis of thin geopolymers. The test specimen was heated in the temperature ranges from 30 °C to 1200 °C with heating rate of 5 °C/min in nitrogen gas atmosphere.

Using Synchrotron radiation X-ray tomographic microscopy (XTM) at SLRI, Thailand, the pore distribution at the fracture surface of heated thin geopolymers was revealed. For a complete dataset, 180° X-ray projections of the sample were captured in 0.1 angular increments. To reduce distortions, polychromatic X-rays were attenuated with aluminum foils with thickness of 200 μm using 14 keV of mean energy. The X-ray projections were collected using a pco.edge 5.5 sCMOS camera with pixel size of 1.44 μm. The data were pre-processed and recreated in three dimensions by Octopus software using a filtered-back projection algorithm. The rebuilt 3D images were visualized using Drishti software.

D2 Phaser, Bruker X-ray Diffractometer was used to identify the phases present in thin geopolymers. The diffractometer was operated at 40 kV and 35 mA using Cu–Kα radiation. The scan range was set at 10–80°2θ with 2°2θ per minute of scan rate. The diffraction patterns were examined using X’pert High Score Plus software which contained with ICDD PDF-2 database.

In order to reveal the functional groups of thin geopolymers, Perkin Elmer Fourier Transform Infrared Spectroscopy (FTIR) RXI spectrometer was performed. The powdered samples were scanned in the ranges of 650–4000 cm^−1^ using 4 cm^−1^ resolutions.

## 3. Results and Discussion

### 3.1. Properties of Unexposed Thin FAS and FAB Geopolymers

Figure 4 illustrates the bulk density, porosity measurement and flexural strength of unexposed thin FAS and FAB geopolymers. FAS geopolymers achieved flexural strength of 7.8 MPa with a bulk density of 2140 kg/m^3^, apparent porosity of 19.3% and water absorption of 10.1%. The increase in the dosage of borax decahydrate reduced the bulk density of thin geopolymers based on FA and LFS. The FAB geopolymers had a lower bulk density of 1993–2000 kg/m^3^ with higher porosity (20.3–21.4%) and water absorption (10.9–11.5%). A looser matrix was produced due to the larger particle size of borax decahydrate (Figure 2) as compared to when FA and LFS were introduced into the geopolymer system. Bagheri et al. [22] reported a drop in bulk density from 1912 kg/m^3^ to 1803 kg/m^3^ in fly ash-based geopolymers when the borax content in replacing the sodium silicate solution increased to 70%.

Lower flexural strength of 5.3–6.6 MPa was reported in FAB geopolymers than in FAS geopolymers. The decrease in strength was associated with the decrease in bulk density [30]. A similar decreasing trend in compressive strength was reported by Antoni et al. [31] and Bagheri et al. [22] with a higher borax dosage used in the synthesis of boroaluminosilicate geopolymers. However, these contradicted other reported works. Nazari et al. [20,21] achieved 64 MPa of compressive strength at 90 days and 9.5 MPa of flexural strength at 28 days in fly ash geopolymers alkali-activated by 0.912 borax to NaOH weight ratio. Antoni et al. [32] produced high calcium-fly ash geopolymers and reported a 6.1% increase in compressive strength by dissolving 1–5 wt.% borax into NaOH solution. The difference in the strength results might be due to the different methods of borax induction and content. In this study, borax decahydrate was used as an additive and added to the geopolymer paste during the mixing process. The larger borax particles acted only as a filling role in which the surface would not participate in the chemical reactions and thus formed a loose geopolymer structure at ambient temperature. Even the addition of borax decahydrate improved the workability of FAB geopolymers, but the high calcium LFS aided the setting of thin geopolymers (1.5–2 h). The short setting time was insufficient for the dissolution of borax, causing the borax particles to remain in the geopolymers. This was proved by the white dots spotted on the surface of the FAB geopolymers (Figure 5). Despite that, the flexural strength increased slightly in the FAB8 geopolymers. This might be due to the formation of a B–O bond as a result of the higher borax content [21]. Nevertheless, the thin geopolymers were further heat-treated to improve the flexural performance of the thin geopolymers.

### 3.2. Elevated Temperature Exposure

Thin FAS, FAB2 and FAB8 geopolymers were exposed to elevated temperature according to the physical and mechanical properties aforementioned to evaluate the thermal performance at high temperatures. The FAB2 and FAB8 geopolymers were selected due to the higher flexural strength obtained in the unexposed specimens (Figure 4b). Moreover, a comparison between the effect of lower and higher borax content on the thermomechanical properties of thin geopolymers could be evaluated. Figure 6 illustrates the physical appearance of thin geopolymers that experienced changes as exposed to elevated temperatures. In general, no obvious degradation such as spalling, cracks, edge loss, or corner loss was observed in all thin geopolymers exposed to high temperatures. The color of both FAS and FAB geopolymers changed from gray to yellowish-beige, then brownish-beige as the heating temperature increased. This was associated with the oxidation of iron compounds from FA and LFS and decomposition of limestone from LFS upon exposure to elevated temperature [33]. A slight curvature of specimens was observed with elevated temperature beyond 1000 °C due to thermal shrinkage. Nevertheless, this did not affect the flexural measurements as the bending of specimens was minimal (<3°). Moreover, the white dots present on the surface of FAB geopolymers disappeared at high temperatures as the borax decahydrate melted and fused into the geopolymer matrix [34].

The bulk density and porosity measurements of the thin FAS and FAB geopolymers at elevated temperatures are illustrated in Figure 7 and Table 2 tabulates the gain and loss of bulk density, mass, and volume of the heat-treated thin geopolymers. The trends for both apparent porosity and water absorption were generally similar and consistent with the bulk density measurements. When exposed to high temperatures, all thin geopolymers experienced changes in bulk density. Little change in bulk density was reported in unexposed geopolymers due to moisture evaporation during the 28-day curing process.

In general, the bulk density of thin geopolymers decreased with increasing temperature. This was associated with the liberation of water vapor from the geopolymer structure [35]. Nevertheless, a decrease in porosity and water absorption was observed as the heating temperature reached 900 °C. The high temperature densified the geopolymer matrix, which was caused by solidifying melt [36], reducing the pore content in thin geopolymers. Furthermore, the density loss for both the thin FAS and FAB2 geopolymers at 1100 °C and 1000 °C, respectively, was relatively small. This was related to the shrinking of the sample due to heat or matrix densification, recompensing the mass loss [37]. However, a drastic increase in bulk density of FAB8 geopolymer was reported at 1000 °C, with a 7.3% gain in density. This was attributed to the thermal shrinkage (13.0% mass loss and 14.3% volume loss) (Table 2) caused by the densification and crystallization of the geopolymer structure, which improved the interparticle connectivity as supported by Rickard et al. [38].

Comparatively, lower density and mass loss was reported in FAB geopolymers than in FAS geopolymers (Table 2). Moreover, the apparent porosity (7.7–19.7%) and water absorption (3.5–10.8%) of FAB geopolymers were lower than FAS geopolymers (10.2–18.8% porosity and 4.9–10.2% water absorption) after elevated temperature exposure. The borax decahydrate melted at high temperature and acted as the fluxing agent and assembled the interparticle, resulting in the formation of a more compact structure. The molten B_2_O_3_ fused into the geopolymer matrix and aided the gelation of sodium aluminosilicate hydrate (N–A–S–H), Calcium Silicate Hydrate (C–S–H) and calcium aluminosilicate hydrate (C–A–S–H) at lower temperature and phase transformation at high temperature. The pore structure of thin geopolymers was consequently refined and hence, reducing the apparent porosity and water absorption of FAB geopolymers after undergoing heat treatment. This was attested by Hernández et al. [39].

Figure 8 depicts the flexural strength and relative flexural strength of heated thin FAS and FAB geopolymers at different elevated temperatures. The relative flexural strength was calculated as the quotient of the flexural value at the elevated temperature (f_T_) to the flexural value of the unexposed specimen (f_RT_). The flexural strength generally increased with increasing elevated temperature. Heat facilitated the geopolymerization reaction in unreacted precursor materials, resulting in gelation of more reaction products of N–A–S–H, C–S–H and C–A–S–H gels [40]. Nevertheless, a slight drop in flexural strength to 6.9 MPa was observed in FAS geopolymers at 300 °C. This was related to the evaporation of absorbed and chemically bound water from the geopolymer matrix [13].

Both FAS and FAB geopolymers depicted drastic increase in flexural strength at high temperatures. The FAS geopolymers had the highest flexural strength of 24.1 MPa at 1100 °C, while the FAB2 and FAB8 geopolymers reported flexural strength of 26.5 MPa and 47.8 MPa, respectively, at 1000 °C. The fluxing behavior of borax decahydrate lowered the glass transition temperature of thin geopolymers. Al Saadi et al. [34] observed the melting point of alkali-activated materials based on waste glass reduced from 850 °C to 700 °C with borax addition. The lowered glass transition temperature helped achieve high flexural strength at lower sintering temperatures as compared to FAS geopolymers. This was further supported by the relative flexural value as shown in Figure 8b. The FAB8 and FAB2 geopolymers reported a significant increase in flexural strength with a factor of 3.6 and 4.0 at lower temperatures of 900 °C and 1000 °C, respectively. In comparison, FAS geopolymers were only able to achieve a significant increase in strength with a factor of 3.1 at 1100 °C. The thermal decomposition of borax decahydrate generated B_2_O_3,_ which acted as a fluxing agent, produced an adhesive medium in assembling the aluminosilicates from precursors to form the compact integration [41], and led to the drastic rise in flexural strength (Figure 9). This was consistent with the porosity measurement in Figure 7. Moreover, the borax flux aided the occurrence of earlier phase transformation. More crystalline phases were produced at lower elevated temperature and further boosted up the flexural strength of thin geopolymers. This was supported by Guo et al. [25] who reported an increment of flexural strength from 2.9 MPa to 18 MPa by using borax decahydrate in ceramizable silicone rubber/halloysite composites sintered at 1000 °C.

In addition, the flexural strength of thin geopolymers did not drop with high-temperature exposure. This contradicted the other reported results as the strength deteriorated exponentially after being subjected to high temperatures [42,43,44,45]. Approximately 80–85% of reduction in flexural strength was observed in geopolymer mortar and concrete at high temperatures beyond 800 °C. The excellent strength retention of high temperature-heated thin geopolymers might be associated with the thickness of merely 10 mm, which contributed to more uniform heating and consequently led to a higher degree of crystallization and vapor pressure dissipation [16]. Furthermore, the inclusion of LFS as precursors yielded the formation of more complex crystalline phases [15]. Formation of the refractory phases in thin geopolymers was favored, resulting in high-strength development. The curving of thin geopolymers at high temperatures also did not affect the flexural measurements, as a more rigid body was formed due to solidifying melt and a higher degree of crystallization [36], recompensing the possible flexural failure caused by the curvature of specimens.

### 3.3. Microstructural Analysis

Figure 10 illustrates the microstructure of FAS and FAB8 geopolymers at different elevated temperatures. The unexposed geopolymers (Figure 10a,a′) had a looser but smooth structure with some coarse pores and unreacted FA particles. C–S–H gels were observed as the results of LFS alkali activation. With the aid of thermal treatment, the geopolymer structure became denser and smoother, because of the increasing geopolymerization extent due to heating. This observation was supported by Pan et al. [14]. Heat facilitated the formation of calcium and aluminosilicate hydrates (C–A–S–H gel and N–(C)–A–S–H gel), densifying the geopolymer structure. As the temperature reached 1000 °C and above, crystalline phases were formed with rigid structures (Figure 10e,e′,f). The formation of crystalline phases contributed to the significant rise in flexural strengths as shown in Figure 8. Murri et al. [15] who also observed the presence of crystalline phases in the structure of FA/LFS geopolymers at 1000 °C, reported an increase in compressive strength from 14 MPa to 42 MPa.

On the other hand, the fluxing behavior of borax decahydrate could be proven via microstructural analysis. The borax addition lowered the glass transition temperature and promoted crystallization at a lower sintering temperature. At 900 °C, FAS geopolymers exhibited a looser structure than FAB8 geopolymers (Figure 10d,d′). A continuous and glassy matrix with partial melting was observed in FAB8 geopolymers, which was known as solidifying melt process. This contributed to viscous flow which filled up the existing voids and pores, increasing the compactness of the microstructure [46]. A similar glassy structure was observed by Al Saadi et al. [34] at 650 °C in alkali-activated boroaluminosilicate based on waste glass. Furthermore, FAB8 geopolymers exhibited more crystalline phases at high temperatures as compared to FAS geopolymers. The incorporation of borax decahydrate promoted the formation of large number of crystalline phases at lower temperature of 1000 °C (Figure 10e′). This was attested by Hernández et al. [39]. The high number of crystalline phases in 1000 °C-exposed FAB8 geopolymers was in parallel with the highest flexural strength achieved in Figure 8 and validated the thermal transformation shown in Figure 9.

### 3.4. Thermogravimetric Analysis

Figure 11 shows the TGA, DTG and DSC curves of FAS and FAB8 geopolymers. Based on TGA/DTG results (Figure 11a,b), a significant mass loss was observed at a temperature below 200 °C. This was attributed to the evaporation of the weakly absorbed and chemically bound water in the geopolymer matrix, as previously reported [35]. Lower mass loss was observed in FAB8 geopolymers in this section, indicating FAB8 geopolymers exhibited lesser water content than FAS geopolymers. After 200 °C, the rate of mass loss reduced, and an intense peak was found at around 600 °C, which was associated with the CaCO_3_ decomposition [33]. There was no noticeable mass change beyond that temperature for both FAS and FAB8 geopolymers, which was commonly reported in previous works [47].

The total mass loss of FAS and FAB8 geopolymers were 8.2% and 6.9%, respectively, which is consistent with the mass loss results tabulated in Table 2. This supposed that the FAB8 geopolymer was more thermally stable than the FAS geopolymers. The molten borax decahydrate fused into the geopolymer matrix and favored the formation of calcium and aluminosilicate hydrate gel [25,41]. The formation of more hydration gels consequently improved the thermal stability of FAB8 geopolymers. In contrast, Hernández et al. [39] reported the mass loss of geopolymers increased with increasing boron content. This might be associated with the usage of boric acid as the fluxing agent, as the decomposition of boric acid contained several stages which contributed to more mass loss.

On the other hand, a downward deformation curve was observed in the DSC curves (Figure 11c). This was attributed to the endo-energetic effect of the continuation of the geopolymerization reaction. The endothermic reaction was in favor of its utilization as fire-resistant materials [48], which indicated the thin geopolymer was suitable to be applied in passive fire protection features. Moreover, the DSC curves showed exothermic effects in both FAS and FAB8 geopolymers at around 1100 °C, which was associated with the structural changes of matrix and crystalline phase formation [28]. This exothermic peak was skewed slightly to the left-hand side in FAB8 geopolymers, proving the glass transition temperature was lowered with borax decahydrate acting as the flux. This validated the fluxing properties of borax as demonstrated in Figure 9.

### 3.5. Pore Distribution Analysis

The pore structures, pore size distribution with pore size percentages and pore content of FAS and FAB8 geopolymers using synchrotron X-ray tomography were illustrated in Figure 12. FAS and FAB8 geopolymers at 1100 °C and 1000 °C, respectively, were selected to evaluate the pore structure due to their high-strength performance. The FAS geopolymers (Figure 12a) had higher porosity than the FAB8 geopolymers (Figure 12b) as more “air” space (purple color) was observed. This complied with the porosity measurement in Figure 7. Moreover, the pore size range of the FAS geopolymers was wider (1–75 μm) than the FAB8 geopolymers (1–55 μm). For instance, 57.22% of the pores in the FAB8 geopolymers fall in the range of 1–5 μm, while 57.86% of the pore size of the FAS geopolymers ranged from 5–20 μm. This indicated that the FAS geopolymers had a larger pore size, resulting in lower flexural strength as larger pores could easily initiate flexural failure.

The FAS geopolymers had a total porosity of 23.6% with 20.1% of open pores and 3.5% of closed pores. In contrast, the porosity of the FAB8 geopolymers was lower (20.4% total porosity, 19.8% open pores and 0.6% closed pores). The borax decahydrate decomposed into molten B_2_O_3_ and lowered the temperature required for crystallization [34]. Thus, more crystalline phases were formed, and the pore structure could be densified at elevated temperature, which was consistent with the SEM micrographs in Figure 10 as the microstructure of the FAB8 geopolymers was more rigid and full of crystalline phases as compared to the FAS geopolymers. This was also supported by Aziz et al. [49] as the higher crystallinity reduced the porosity of 1200 °C heat-treated alkali-activated slag.

### 3.6. Phase Analysis

Figure 13 displays the XRD diffractograms of FA, LFS and borax decahydrate. The XRD patterns demonstrate that FA contained mineral phases of quartz (SiO_2_) and mullite (3Al_2_O_3_∙2SiO_2_) with minor phases of hematite (Fe_2_O_3_). A broad hump was observed in FA from 15°–35° (2θ), indicating the amorphous characteristics of the materials. The primary phases found in LFS were calcio-olivine (Ca_2_SiO_4_) with glassy phases of merwinite (Ca_3_MgSi_2_O_8_), magnetite (Fe_3_O_4_) and calcium aluminum oxide (CaAl_2_O_4_). For borax decahydrate, only intense peaks of sodium tetraborate decahydrate (Na_2_B_4_O_7_·10H_2_O) were detected.

The XRD diffractograms of the FAS and FAB8 geopolymers at different elevated temperatures are illustrated in Figure 14. The position of the diffuse halo was shifted to a higher degree of 20°–40° (2θ), indicating the formation of amorphous aluminosilicate hydrate gel N–A–S–H after geopolymerization reaction [50]. Moreover, phases of quartz and mullite originating from FA remained in both the FAS and FAB8 geopolymers, denoting that the quartz and mullite did not fully participate in the reaction. The formation of crystalline peaks of calcite (CaCO_3_) and calcium silicate hydrate (C–S–H) were the results of LFS alkali activation. This complied with the SEM micrographs of unexposed geopolymers in Figure 10a,a′ as unreacted FA particles and C–S–H gels were observed in the geopolymer structure.

The development of stable crystalline phases was facilitated by high-temperature exposure. The broad hump’s intensity reduced, resulting in a decrease in the amorphous content of geopolymers decreased after being subjected to high temperature. The heat applied triggered the eutectic reactions within SiO_2_, Al_2_O_3_, Na_2_O and CaO from the precursors, forming new crystalline phases of anorthite (CaAl_2_Si_2_O_8_), albite (NaAlSi_3_O_8_) and nepheline (NaAlSiO_4_) in both FAS and FAB8 geopolymers. The presence of borax decahydrate in FAB8 geopolymers decomposed into molten B_2_O_3_ and facilitated the phase transformation with more crystal formation. The decomposition of borax decahydrates and eutectic crystallization reaction from the SiO_2_, Al_2_O_3_, Na_2_O and CaO are shown in Equations (3)–(7) [25,51,52].

Decomposition of borax decahydrate:70~100 °C Na_2_B_4_O_7_·10H_2_O → Na_2_B_4_O_7_ + 10H_2_O(3)
100~500 °C Na_2_B_4_O_7_ → 2NaBO_2_ + B_2_O_3_ (molten boron)(4)

Eutectic crystallisation reactions:2SiO_2_ + Al_2_O_3_ + CaO → CaAl_2_Si_2_O_8_ (anorthite)(5)
6SiO_2_ + Al_2_O_3_ + Na_2_O → 2NaAlSi_3_O_8_ (albite)(6)
2SiO_2_ + Al_2_O_3_ + Na_2_O → 2NaAlSiO_4_ (nepheline)(7)

The molten boron acted as a fluxing agent and improved the interconnectivity of the particles, increasing the degree of crystallization of thin geopolymers. More refractory crystalline phases were detected in FAB8 geopolymers at a lower sintering temperature (Figure 14b). The higher number of these refractory crystals present in the dense and glassy matrix enhanced the flexural strength, which was attested by Skvara et al. [53]. In addition, the crystalline phases could act as reinforcement and improve the rigidity of the geopolymer structure [54]. Consequently, a significant increase in flexural strength in the FAB8 geopolymers beyond 900 °C was achieved as compared to the FAS geopolymers. Thus, this showed that the fluxing behavior of borax decahydrate could lower the glass transition temperature and further enhance the flexural properties of thin geopolymers.

### 3.7. Functional Group Identification

The FTIR spectra of FA, LFS and borax decahydrate are illustrated in Figure 15. The main broad band of FA at 1032 cm^−1^, was related to the asymmetrical Si–O–Si and Si–O–Al stretching vibration [55]. A minor band at 775 cm^−1^ corresponds to the symmetrical vibrations of Si–O or Al–O [56]. On the other hand, the main band of LFS at 858 cm^−1^ was the bending vibration of Ca–O and Si–O [57]. The O–C–O stretching vibration was identified as bands located at 2168 cm^−1^, 2025 cm^−1^ and 1419 cm^−1^ [11]. Borax decahydrate displayed characteristic peaks of Na_2_B_4_O_7_·10H_2_O as reported in Goel et al. [58]. The absorption bands at 1363 cm^−1^, were claimed as the B–O asymmetric stretching in BO_3_. The bands at 1008 cm^−1^ and 813 cm^−1^ were the asymmetric and symmetric stretching of the B–O bond in BO_4_, respectively. Moreover, the band at 1651 cm^−1^ and the broad bands at 3172 cm^−1^ and 3312 cm^−1^ were traces of H–O–H bending and –OH stretching vibrations.

Figure 16 displays the FTIR spectra of the FAS and FAB8 geopolymers at elevated temperatures. For unexposed specimens, both the FAS and FAB8 geopolymers showed shifting in functional groups after being alkali-activated. The shifting of the FA main broad band to a lower wavenumber of ~970 cm^−1^ certified the formation of amorphous N–A–S–H gel [55]. Furthermore, the main band of LFS also shifted to ~870 cm^−1^, verifying the formation of C–A–S–H gel [59]. For the FAB8 geopolymer, a minor band at 1381 cm^−1^ was observed, implying the B–O bond formation due to the addition of borax decahydrate [20]. Nevertheless, the bands for the O–C–O stretching vibration (~1420 cm^−1^ and ~2020 cm^−1^) were still present and the bands for the H–O–H bending and –OH stretching vibrations were detected at ~3300 cm^−1^ and ~1650 cm^−1^.

On the other hand, no new additional band was observed in heated thin geopolymers. In fact, the band for the bending vibrations of Ca–O and Si–O of both the FAS and FAB8 geopolymers disappeared at high temperatures. This might be attributed to the decomposition of calcite and formation of anorthite at high temperature [49], which was consistent with the TGA and XRD results in Figure 11a and Figure 14, respectively. Moreover, the main band for the asymmetrical stretching vibration of Si–O–Si and Si–O–Al broadened at elevated temperatures. This was associated with the structural disorder, which was in parallel with previous studies [10,60]. Moreover, an increase in transmittance percentage of the bands at ~3300 cm^−1^ and ~1650 cm^−1^ was observed, demonstrating the decrease in the intensity of H–O–H and –OH bonding, and proving the full dehydration of geopolymers at high temperatures.

## 4. Conclusions

In this article, the effect of borax decahydrate addition on the flexural properties of thin geopolymers based on FA and LFS at elevated temperatures was explored. Unexposed FAB geopolymers experienced a slight drop in flexural strength to 5.3–6.6 MPa as compared to FAS geopolymers (7.8 MPa). A looser structure with higher porosity led by the addition of larger borax particle size obstructed the flexural performance of the FAB geopolymers. However, elevated temperature exposure stimulated the fluxing properties of borax decahydrate and improved the flexural properties of the thin geopolymers. Heat treatment boosted the thin FAS geopolymers up to 24.1 MPa at 1100 °C. With the aid of borax addition, the molten B_2_O_3_ which acted as a flux, produced an adhesive medium to assemble the aluminosilicates and thus formed a compact integration. This caused a drastic rise in flexural strength where the FAB2 and FAB8 geopolymers reported a flexural strength of 26.5 MPa and 47.8 MPa, respectively, at 1000 °C.

Moreover, this paper also demonstrated that the glass transition temperature of thin FA/LFS geopolymers could be reduced via borax addition. FAB geopolymers were able to achieve a significant increase in strength with a factor of 3.6 for FAB8 and 4.0 for FAB2 at lower temperatures of 900 °C and 1000 °C, respectively. Comparatively, FAS geopolymers only achieved a 3.1 factor in strength increment at a higher temperature of 1100 °C. With the lowered glass transition temperature, the degree of the eutectic crystallization reaction in FAB geopolymers was facilitated, forming more refractory crystalline phases at lower sintering temperatures as compared to the FAS geopolymers. Crystalline phases of anorthite, albite and nepheline were formed, strengthening the geopolymer structure and hence improving the flexural properties. As the degree of crystallization was increased in FAB geopolymers, more refractory crystals were detected in the dense and glassy matrix. The pore structure was refined as the interparticle connectivity was improved via the reinforcement by crystalline phases, hence enhancing the flexural properties of the thin geopolymers. The achievement of high flexural strength in this study consequently validates the incorporation of borax decahydrate in inducing the high strength development of thin geopolymers.

## Figures and Tables

**Figure 1 materials-15-04178-f001:**
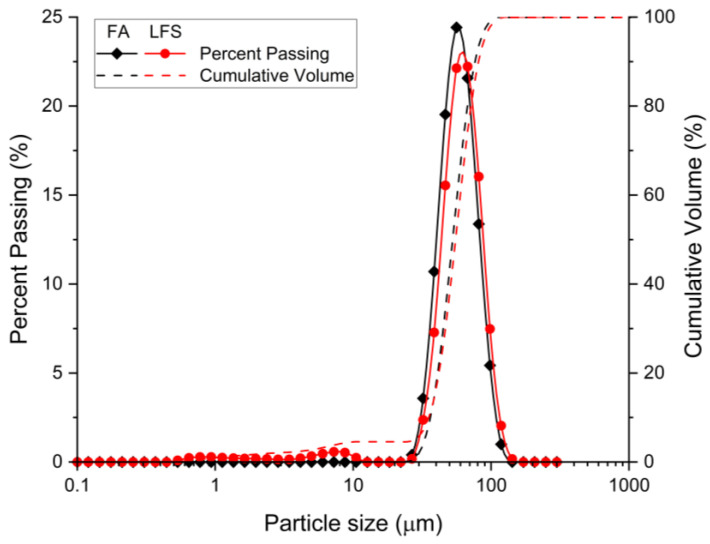
Particle size distribution of FA and LFS.

**Figure 2 materials-15-04178-f002:**
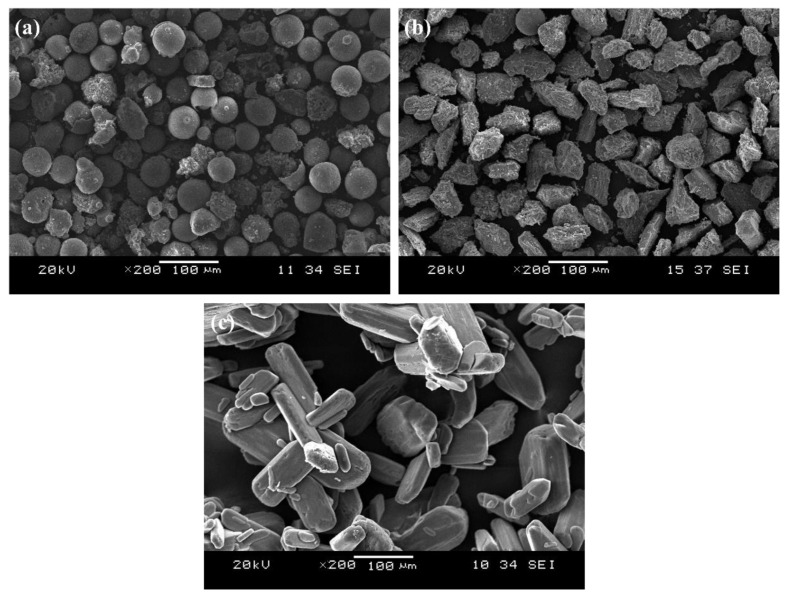
SEM micrographs of (**a**) FA, (**b**) LFS, and (**c**) borax decahydrate.

**Figure 3 materials-15-04178-f003:**
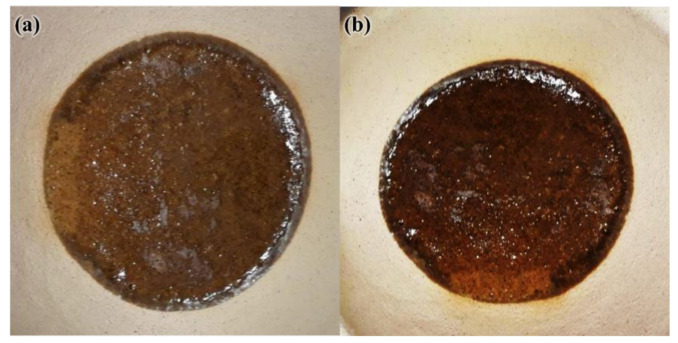
Heated thin (**a**) FAS geopolymer and (**b**) FAB geopolymer at temperature above 1100 °C and 1000 °C, respectively.

**Figure 4 materials-15-04178-f004:**
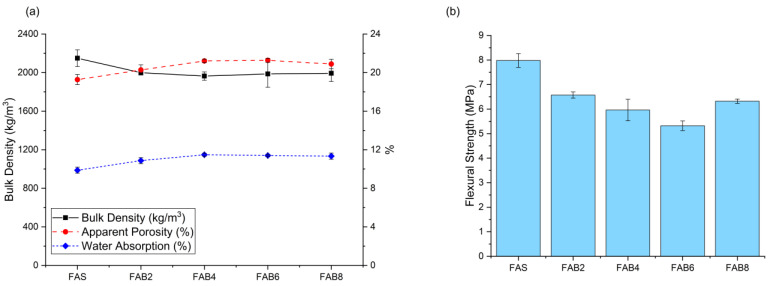
(**a**) Bulk density and porosity measurement and (**b**) flexural strength of unexposed thin FAS and FAB geopolymers.

**Figure 5 materials-15-04178-f005:**
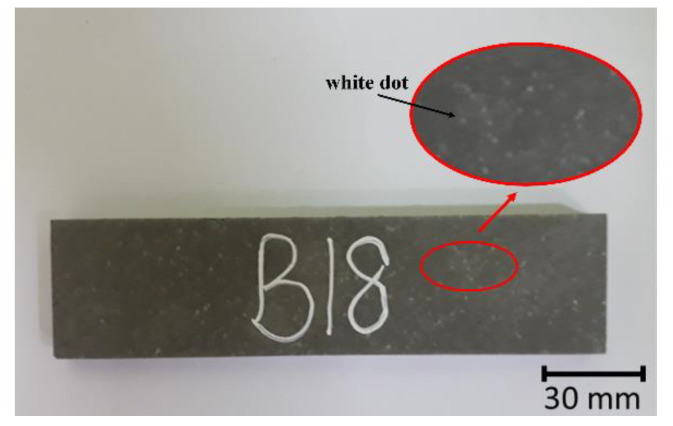
Surface condition of FAB geopolymer with magnified view of white dots in the red circle area.

**Figure 6 materials-15-04178-f006:**
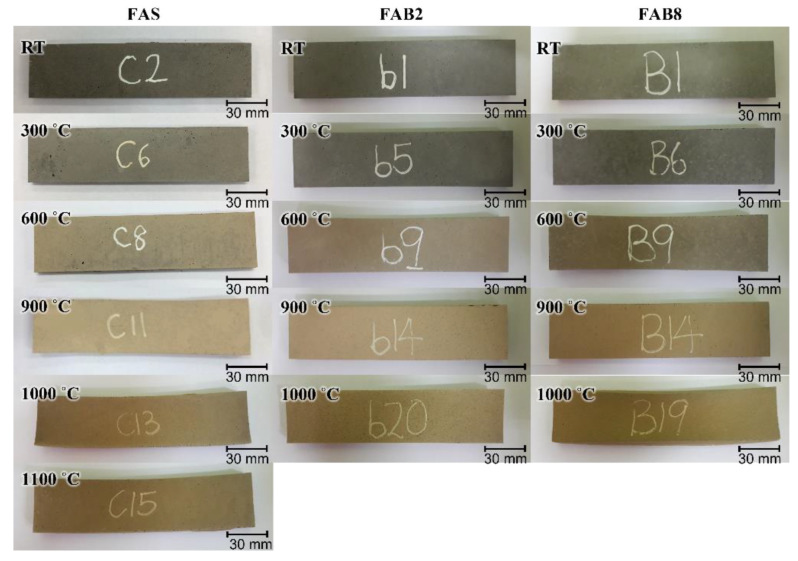
Physical evolution of unheated and heated thin geopolymers at 300 °C, 600 °C, 900 °C, 1000 °C and 1100 °C.

**Figure 7 materials-15-04178-f007:**
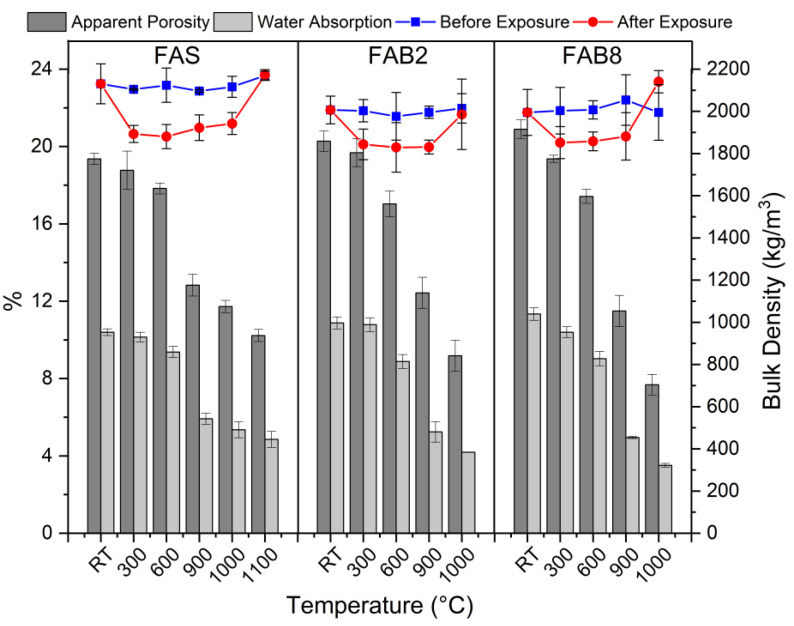
Bulk density, apparent porosity, and water absorption of thin FAS and FAB geopolymers at elevated temperatures.

**Figure 8 materials-15-04178-f008:**
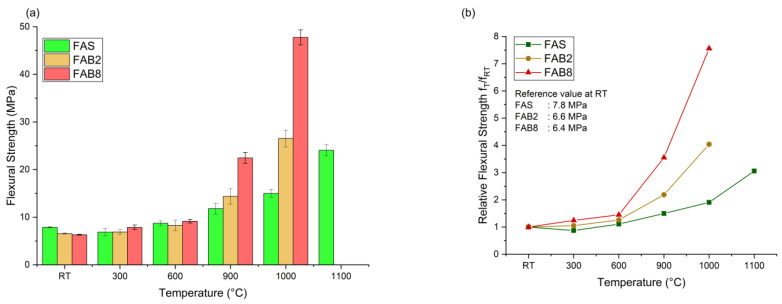
(**a**) Flexural strength and (**b**) relative flexural strength of FAS and FAB geopolymers at different elevated temperatures.

**Figure 9 materials-15-04178-f009:**
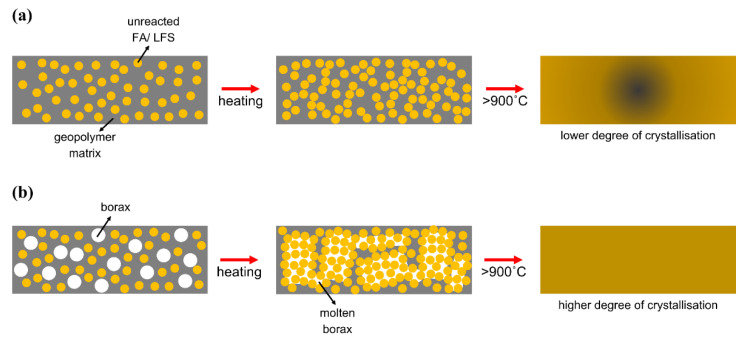
Schematic diagram of thermal transformation of thin geopolymers (**a**) without and (**b**) with borax decahydrate addition.

**Figure 10 materials-15-04178-f010:**
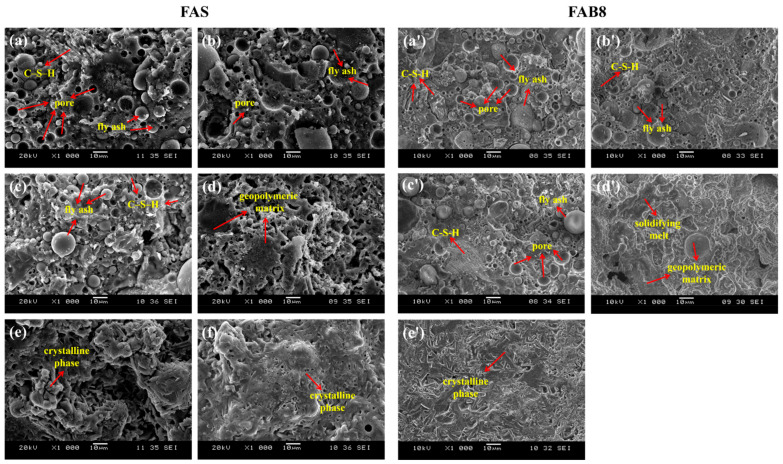
SEM micrographs of FAS (**left**) and FAB8 (**right**) geopolymers exposed to elevated temperature of (**a**,**a**′) RT; (**b**,**b**′) 300 °C; (**c**,**c**′) 600 °C; (**d**,**d**′) 900 °C; (**e**,**e**′) 1000 °C and (**f**) 1100 °C.

**Figure 11 materials-15-04178-f011:**
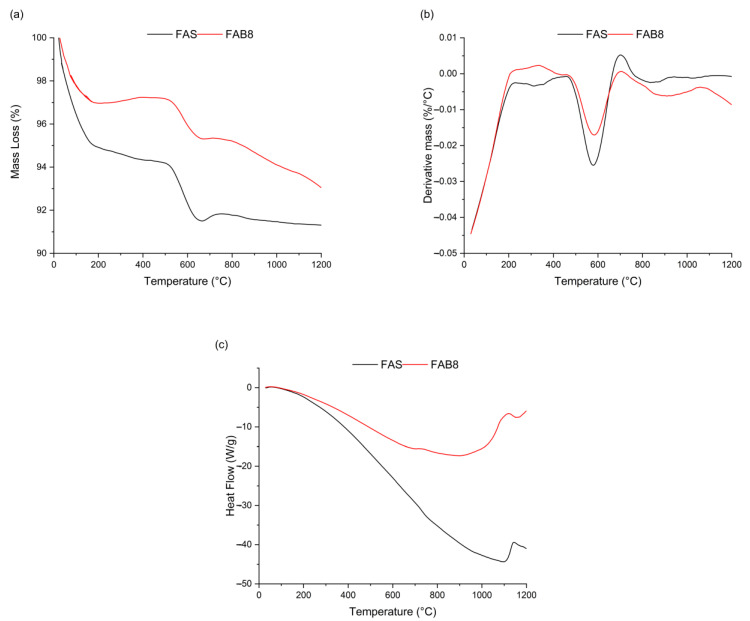
(**a**) TGA, (**b**) DTG and (**c**) DSC curves of FAS and FAB8 geopolymers.

**Figure 12 materials-15-04178-f012:**
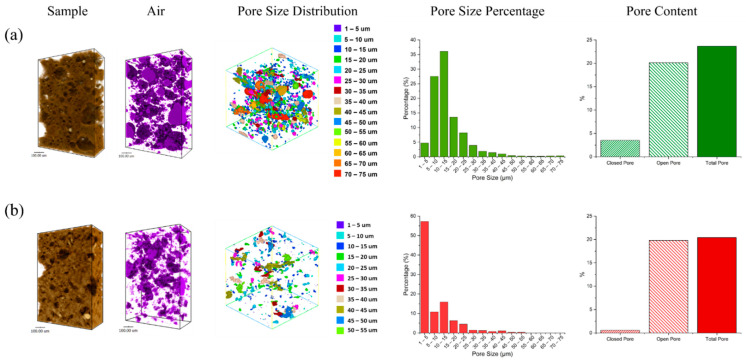
Pore structures with pore size distributions of (**a**) FAS geopolymer at 1100 °C, (**b**) FAB8 geopolymer at 1000 °C.

**Figure 13 materials-15-04178-f013:**
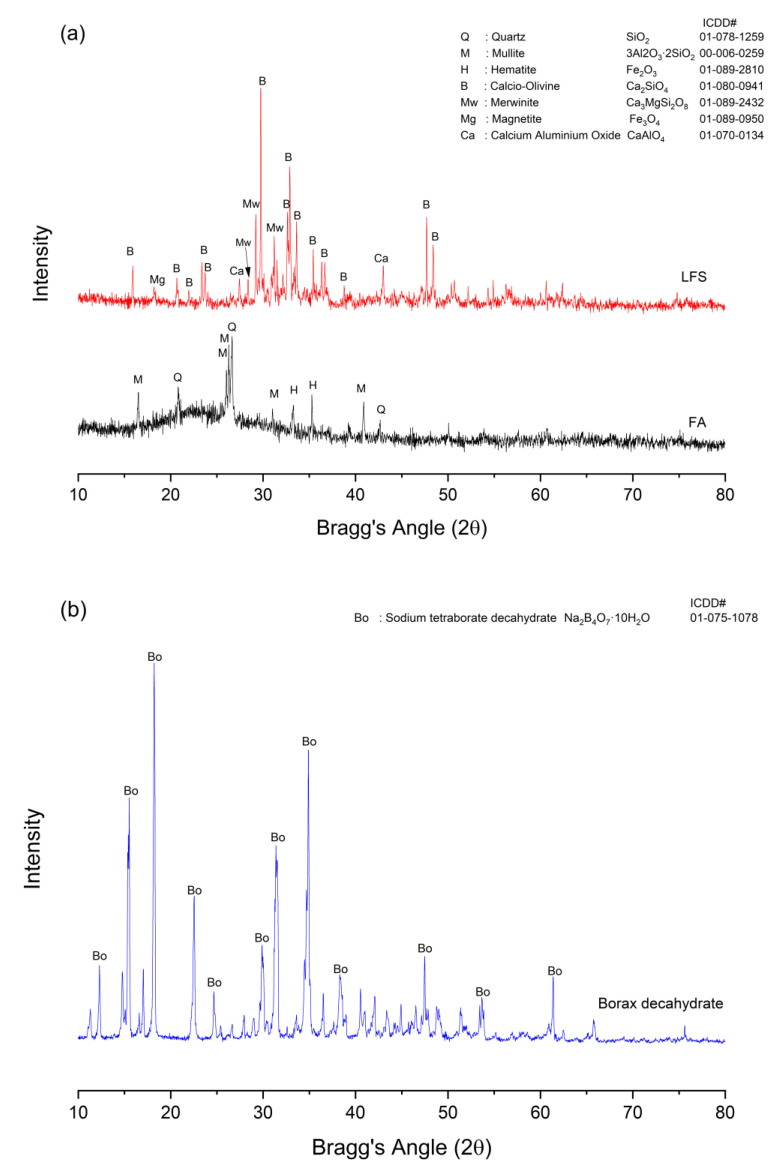
XRD diffractograms of (**a**) FA, LFS and (**b**) borax decahydrate.

**Figure 14 materials-15-04178-f014:**
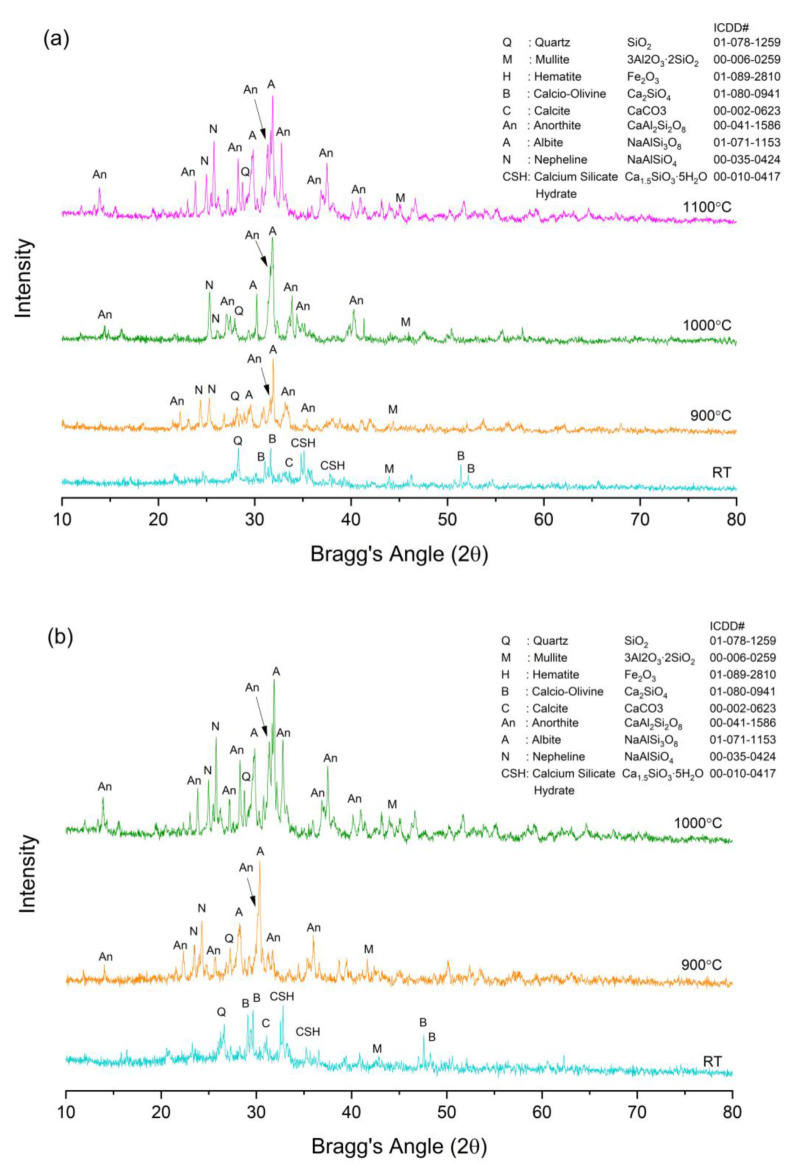
XRD diffractograms of unexposed and exposed (**a**) FAS and (**b**) FAB8 geopolymers.

**Figure 15 materials-15-04178-f015:**
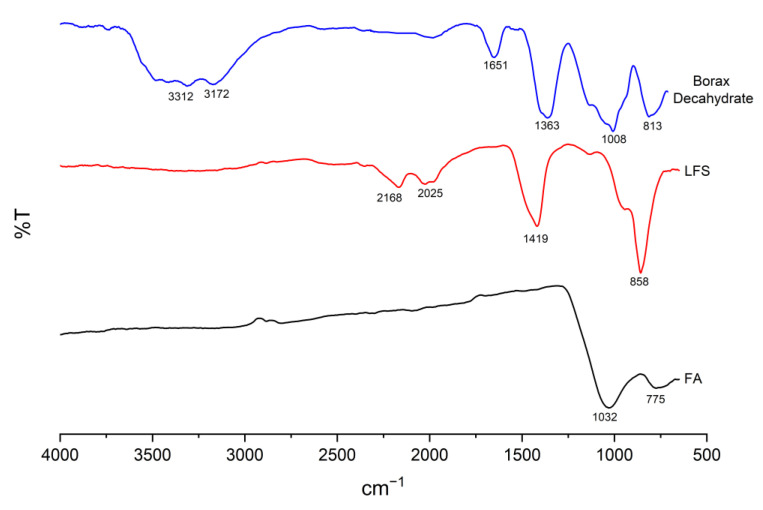
FTIR spectra of raw FA, LFS and borax decahydrate.

**Figure 16 materials-15-04178-f016:**
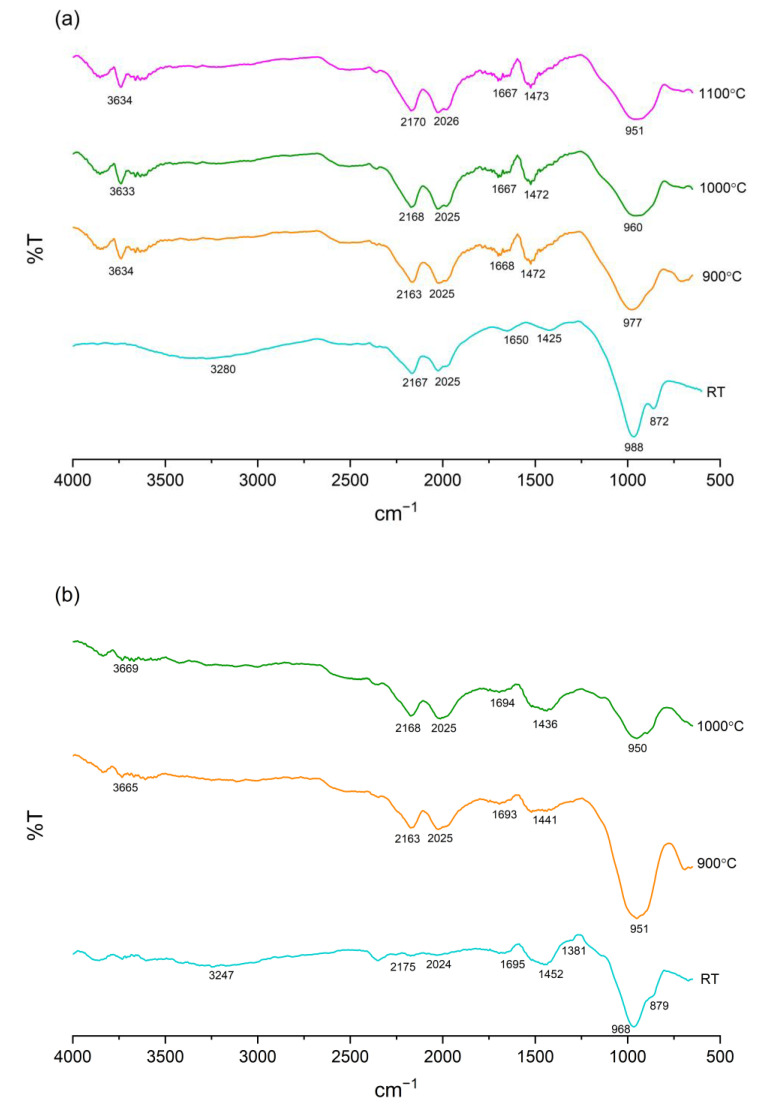
FTIR spectra of (**a**) FAS and (**b**) FAB8 geopolymers at elevated temperatures.

**Table 1 materials-15-04178-t001:** Chemical composition and loss on ignition of fly ash and ladle furnace slag.

Compound	SiO_2_	Al_2_O_3_	CaO	Fe_2_O_3_	MgO	TiO_2_	K_2_O	Others	LOI (%)
FA (wt.%)	56.30	28.00	3.89	6.86	-	2.17	1.49	1.29	1.95
LFS (wt.%)	21.30	2.30	63.59	8.08	2.60	0.50	-	1.63	3.74

**Table 2 materials-15-04178-t002:** Gain and loss of bulk density, mass, and volume of thin FAS and FAB geopolymers at elevated temperatures.

	FAS (%)			FAB2 (%)			FAB8 (%)		
	Density Gain/Loss	Mass Gain/Loss	Volume Gain/Loss	Density Gain/Loss	Mass Gain/Loss	Volume Gain/Loss	Density Gain/Loss	Mass Gain/Loss	Volume Gain/Loss
RT	−0.1	−0.08	−0.09	−0.06	−0.07	−0.1	−0.08	−0.08	−0.09
300 °C	−10.0	−10.2	−7.8	−8.0	−9.9	−5.6	−7.6	−9.7	−4.2
600 °C	−11.5	−12.8	−8.7	−7.4	−12.5	−6.4	−7.4	−11.8	−3.4
900 °C	−8.3	−13.8	+5.5	−8.4	−13.7	+5.4	−8.3	−13.4	+5.2
1000 °C	−8.2	−14.1	+4.9	−1.4	−13.5	−7.9	+7.3	−13.0	−14.3
1100 °C	+0.1	−14.0	−8.2	-	-	-	-	-	-

## Data Availability

All the data is available within the manuscript.

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
