# Peer review of "Improvements of Flexural Properties and Thermal Performance in Thin Geopolymer Based on Fly Ash and Ladle Furnace Slag Using Borax Decahydrates"

_materials, 2022, doi:10.3390/ma15124178_

Round 1
Author Response
Comment 1: Please redraw Figure 1 to make it easier to distinguish between the two different materials. For example, add the small scatter plot shape.
Response: Figure 1 has been redrawn and small scatter plot was added to better distinguish the particle size distribution of fly ash and ladle furnace slag (Page 3 / Figure 1).
Comment 2: The introduction must be improved. Some latest reference like: Gaili Xue, Erol Yilmaz, Guorui Feng, Shuai Cao and Lijuan Sun, Reinforcement effect of polypropylene fiber on dynamic properties of cemented tailings backfill under SHPB impact loading, Construction and Building Materials, 279(2021), No.122417 should be added.
Response: The introduction was improved, and latest references have been updated (Page 2 / Lines 63 – 70).
Comment 3: Line 149, the basis for setting the temperature?
Response: The elevated temperatures were set based on previous literatures [1] as the thermo-physical and chemical changes of geopolymers such as further geopolymerisation, dehydration and crystallisation of geopolymers could be observed within these temperature ranges. The upper boundary of heating temperature for FAS and FAB geopolymers were fixed at 1100°C and 1000°C, respectively, as the thin geopolymers would melt beyond these temperatures. The explanation has been added (Page 4 – 5 / Lines 152 – 155).
Comment 4: Line 219, the large borax particles only play a filling role, because the particle size is large, then the surface will not participate in chemical reactions.
Response: The sentence has been rephrased (Page 6 – 7 / Lines 224 – 226).
Comment 5: Figure 8 shows the schematic diagram, are there any further test studies?
Response: The schematic diagram in Figure 9 was further validated by the SEM analysis (Page 11 / Figure 10), thermogravimetric analysis (Page 12 / Figure 11) and pore distribution analysis (Page 13 / Figure 12) which demonstrated that the fluxing properties of borax as the glass transition temperature of thin geopolymers was lowered, facilitating the degree of crystallisation and thus formed a more rigid body with large number of crystalline phases.
Reviewer 2 Report
Dear authors, please find attached my comments.

Author Response
Manuscript Number: materials-1742972
Paper Title: Improvements of Flexural Properties and Thermal Performance in Thin Geopolymer based on Fly Ash and Ladle Furnace Slag using Borax Decahydrates
Dear Mr. Mircea Filipescu,
Thank you for your email dated 23 May 2022. Also, the authors would like to express their appreciation to the reviewers for their precious time and comments. We have carefully addressed the comments below. The replies and the changes have been made in the new version of manuscript. The corrections are highlighted in the manuscript.
Thank you. We look forward your positive response.
Yours Sincerely,
Liew Yun-Ming
Comment 1: Table 1: The LOI must be included.
Response: The loss on ignition (LOI) of fly ash and ladle furnace slag was added in Table 1 (Page 3).
Comment 2: Line 152: …above these temperatures… ?
Response: The sentence has been rephrased (Page 5 / Line 157).
Comment 3: Line 206: After this figure the numbering of the figures must be corrected in both captions and text.
Response: We are sorry for the mistakes made. All the numbering of figures were corrected in both captions and text.
Comment 4: Line 223: Please rephrase the sentence.
Response: The sentence has been rephrased (Page 7 / Line 231 – 232).
Comment 5: Line 242: What about the curving of specimens when exposed to elevated temperatures? Has to do only with the geometry of the specimens? Does this process affect the results of flexural measurements? Please explain.
Response: Elevated temperature exposure caused the thin geopolymers curved slightly above 1000°C due to thermal shrinkage. However, the flexural measurements were not affected as the specimen curvature was minimal (<3°) (Page 7 / Line 251 – 253). Besides, the flexural failure was recompensed by the rigid structure formed caused by solidifying melt and formation of refractory crystalline phases [2] (Page 10 / Line 338 – 341). The explanation has been added.
Comment 6: Line 302: …are only… ?
Response: The sentence has been rephrased (Page 10 / Line 318).
Comment 7: Line 358: It is proposed to add the DTG curves on TGA analysis. In this way, the mass loss steps will be more distinct while the % mass loss of each step can be easily exported.
Response: The DTG curves were added in Figure 11 (Page 12).
Comment 8: Line 373: …contrast result…
Response: The sentence has been rephrased (Page 13 / Line 396 – 397).
Comment 9: Line 381: …which was be associated…
Response: The sentence has been rephrased (Page 13 / Line 405).
Comment 10: The authors have to thoroughly justify the selection of geopolymers containing 2 and 8 % Borax for the analysis at elevated temperatures.
Response: The selection of FAB2 and FAB8 geopolymers for elevated temperature exposure was based on the physical and mechanical properties observed in their unexposed specimens. FAB2 and FAB8 geopolymers were heated due to higher flexural strength obtained compared to FAB4 and FAB6 geopolymers. Besides, comparison between the effect of lower and higher borax content on the thermal performance on FAB geopolymers could be made. The explanation was added (Page 7 / Line 239 – 244).